# Changes in Amino Acid Metabolism of *Staphylococcus aureus* following Growth to the Stationary Phase under Adjusted Growth Conditions

**DOI:** 10.3390/microorganisms10081503

**Published:** 2022-07-25

**Authors:** Mousa Alreshidi, Hugh Dunstan, Tim Roberts, Fevzi Bardakci, Riadh Badraoui, Mohd Adnan, Mohd Saeed, Fayez Alreshidi, Yazeed Albulaihed, Mejdi Snoussi

**Affiliations:** 1Department of Biology, College of Science, University of Ha’il, Hail P.O. Box 2440, Saudi Arabia; fbardakci@adu.edu.tr (F.B.); riadh.badraoui@fmt.utm.tn (R.B.); mo.adnan@uoh.edu.sa (M.A.); mo.saeed@uoh.edu.sa (M.S.); y_albulaihed@yahoo.com (Y.A.); m.snoussi@uoh.edu.sa (M.S.); 2Molecular Diagnostic and Personalized Therapeutics Unit, University of Ha’il, Hail P.O. Box 2440, Saudi Arabia; 3InnovAAte Pty Ltd., 45 Hunter Street, Newcastle, NSW 2300, Australia; hugh@innovaate.com.au; 4Metabolic Research Group, Faculty of Science, School of Environmental and Life Sciences, University Drive, Callaghan, NSW 2308, Australia; tim.roberts@newcastle.edu.au; 5Section of Histology-Cytology, Medicine Faculty of Tunis, University of Tunis El Manar, La Rabta, Tunis 1017, Tunisia; 6Department of Family and Community Medicine, College of Medicine, University of Ha’il, Hail P.O. Box 2440, Saudi Arabia; fs.alreshidi@uoh.edu.sa; 7Laboratory of Genetics, Biodiversity and Valorisation of Bioresources, High Institute of Biotechnology, University of Monastir, Monastir 5000, Tunisia

**Keywords:** *Staphylococcus aureus*, amino acid metabolism, adaptation

## Abstract

The sharp increase in infections due to *Staphylococcus aureus* is associated with its ability to adapt to changes in its habitat. This study aimed to investigate the differences in the cytoplasmic amino acid profiles of a clinical strain of *S. aureus* under five combinations of stress-induced conditions representative of a wound site by varying temperature 35–37 °C, adding 0–5% NaCl and adjusting pH 6–8. The results indicated that aspartic acid, lysine, glutamic acid and histidine were the most abundant cytoplasmic amino acids in the control samples grown under optimal growth conditions. However, the magnitudes and levels of these amino acids were altered under the various wound site conditions, which led to differential cytoplasmic amino acid profiles as characterized by multivariate analyses (PLS-DA). The total cytoplasmic amino acid content was significantly reduced in the cells grown with 2.5% NaCl added at pH 7 and 37 °C relative to the control samples and other growth regimes. However, all combinations of enhanced stress conditions showed unique and characteristic changes in the concentration profiles of the cytoplasmic amino acids. These outcomes supported the hypothesis that bacterial cells of *S. aureus* maintain different metabolic homeostasis under various stress-induced conditions. The potent capability of *S. aureus* to constantly and rapidly acclimatize to variations within the environment may reflect the crucial feature supporting its virulence as an opportunistic pathogenic bacterium to invade the wound site. Understanding the control systems governing these marked changes in amino acids during the adaptation to the potential wound site conditions of this dangerous bacterium may offer new clinical controls to combat infection.

## 1. Introduction

*Staphylococcus aureus* induces an extensive array of infections in both hospital and community sites. The ability of *S. aureus* to cause a high number of diseases is mainly attributed to its ability to alter its metabolism as a process of adaptation to external environments and antibiotics exposure [1,2,3,4] as well as its remarkable fitness to defend the immune system in the host, leading to complicated infections [5,6,7]. However, the precise physiological adaptation mechanisms of *Staphylococci* species to limited nutrients and stressful conditions representative of the human wound site have not been extensively studied. It has been suggested that alterations in the profiles of metabolites are crucial for bacterial acclimatization and infection origination [4,8,9]. Microbes are mainly present in the stationary phase in their common habitats where nutrients are limited and stresses present [10,11]. All these events produced significant changes at the metabolic levels [12]. Such changes were similar to those observed during the stationary phase in laboratory studies [10,13]. Therefore, determining how *S. aureus* adapts its amino acid metabolites to starvation and stressful conditions present in the stationary phase and wound site conditions is an important step that may lead to improved clinical practices for preventing the transmission of *S. aureus* infections within hospitals. A wound site is a good model of varied environmental conditions where the pH value changes based on the status of the wound, over the course of the infection becoming neutral or acidic but remaining elevated during the healing process. Levels of NaCl can also be elevated following exposure to traumatization [14,15], and the temperature of the skin can be either decreased or increased between 2–4 degrees relative to the actual temperature based on the wound location and surrounding conditions [16,17]. It has been indicated that staphylococci were the major bacteria found in the wound site in particular the non-healing wounds [18,19]. This result can be interpreted as that *S. aureus* may have a role in delaying the healing time.

Amino acids are crucial components for supporting cell integrity, and their production can be optimized to achieve optimal metabolic homeostasis. They are also important precursors of proteins, nucleotides, lipids, and components in cell wall biosynthesis. It was hypothesized that *S. aureus* induces significant changes in its amino acid compositions to obtain optimal metabolic homeostasis and thus enhance survivability during exposure to stressful conditions present in the stationary phase and during exposure to combinations of environmental parameters representative of a human wound site, including variations in temperature, pH and osmotic pressure. This study aimed to explore the variations in amino acid compositions following the growth of bacterial cells to stationary phase under variations in temperature, pH and sodium chloride (NaCl) representative of a wound site.

## 2. Materials and Methods

### 2.1. S. aureus Strains

*S. aureus* used in this study is a clinical strain collected from a hospital as described earlier [4]. The bacterial strain used in the current study was previously used in the following investigations [4,9,20,21,22,23]. The *S. aureus* strains were identified using API Staph biochemistry and confirmed using 16S rRNA gene primers [24].

### 2.2. Bacterial Growth Conditions and Experimental Design

A starter culture was prepared by inoculating a single colony of *S. aureus* in 100 mL tryptic soy broth (TSB) and incubating at 37 °C overnight for 18 h.

*S. aureus* was grown in TSB in a combination of environmental parameters that mimic the human wound including adjustments in temperature, pH and osmotic pressure implemented by the addition of sodium chloride (NaCl) for investigating cytoplasmic amino acid compositions. The samples represented control samples were cultured under optimal conditions of pH 7 at 37 °C with no added NaCl and five sets of different treatments were implemented with (1) pH 7 at 37 °C with NaCl added (2.5%) (2) 35 °C and pH6 with no added NaCl; (3) 35 °C and pH 8 with no added NaCl; (4) 35 °C and pH 6 with 5% NaCl; (5) 35 °C pH 8 with 5% NaCl added.

The experimental cultures were prepared by transferring five ml of overnight culture to 95 mL of TSB culture media (OD 600 = 0.1 ≈ 108 CFU/mL) and grown to a stationary phase with designated growth conditions prior to the analysis of amino acid compositions. Growth was monitored by aseptically checking the absorbance at 600 nm to estimate cell numbers so that the cultures could be harvested at the stationary phase. Cultures were harvested by centrifugation at 6000× *g* for 20 min followed by washing three times with phosphate buffer saline (PBS). The washed cells were directly quenched using liquid nitrogen and placed in a freezer-dryer machine following extraction for metabolic profiling analysis. The experiments of control and treatment 1) pH 7 at 37 °C with NaCl added (2.5%) were repeated two separate times to optimize statistical assessment and ensure the reproducibility of the experiments.

### 2.3. Analyses of Cell Extracts for Metabolic Profiling

Cytoplasmic metabolites were extracted from lyophilized bacterial cells using a cold methanol/water protocol [25]. Lyophilized cells (≈10–12 mg) were suspended with extraction buffer (1:1 ice-cold methanol/water), mixed, immersed in liquid nitrogen and kept at −20 °C for half an hour to allow the solution to slowly thaw. The contents were then separated by centrifugation to remove cell debris. The supernatant containing the metabolites was placed into new tubes and dried in a drier vacuum to remove the methanol/water. Cytoplasmic amino acids were subsequently extracted and processed with an analytical kit (Phenomenex EZ: faast) for the analyses by Gas Chromatography (GC) as previously described [4].

### 2.4. Processing and Analysis of Metabolic Profile Data

GC-FID data collected from both control and treatment samples were processed using MetaboAnalyst 5.0 and a statistical package (Statistica, TIBCO Software Inc. [2017], data analysis software system, version 13). The data was pre-processed using sum normalization, log transformation, and auto-scaling methods. The statistical significance of amino acids between control and treatment samples was determined using Duncan analysis. Multivariate analyses such as unsupervised principal component analysis (PCA) and partial least squares-discriminant analysis (PLS-DA) were conducted to better understand the subtle common differences between the mid-exponential and stationary samples. Cross-validation including R2 and Q2 values were used to evaluate the goodness of fit and predictive ability of the PLS-DA model. Significant variables that contributed to the separation of various growth stages in the PLS-DA were identified using the variable importance in the projection (VIP) score. The Venn diagram was created by the InteractiVenn website [26].

## 3. Results

The amino acid profiles for the *S. aureus* cells were reproducible within treatment groups, but each group showed significant differences in concentrations for certain amino acids between the various control and stress-induced cultures (Table 1). Following an investigation of the levels of the amino acids in the control samples (n = 6), aspartic acid, lysine, glutamic acid and histidine were found to be the most abundant amino acids in the cytoplasm, accounting for 44.6%, 16.6%, 12.9%, and 9.3%, respectively, of the total amino acids (Table 1). When *S. aureus* cells were exposed to altered environmental conditions, designated Treatments 1–5 in Table 1, aspartic acid continued to be the major amino acid under Treatments 1, 2, and 5, but glutamic acid was the major amino acid in Treatments 3 and 4, and its levels significantly increased under Treatments 2–5. Under all treatments, lysine was substantially lower, and histidine was also significantly reduced except in Treatment 3, where it showed no statistically significant changes. Several specific changes in amino acid levels associated with different treatments were apparent, as shown in Table 1. The total average amino acid yield for the control samples was 407.01 ± 47.9 nmol mg-1 (mean ± SD), but it was significantly reduced in the bacterial cells exposed to 2.5 NaCl at pH 7 and 37 °C (Treatment 1). This reduction was mainly accomplished by 30–50% declines in histidine and lysine. Other treatments showed no significant changes in their total amino acids relative to the control samples, but the relative abundances of individual amino acids were significantly different in the treatments in comparison with the control, as noted in Table 1.

The growth of *S. aureus* at a lower temperature of 35 °C with no addition of sodium chloride at pH 6 (Treatment 2) resulted in extraordinarily different responses within individual amino acids in contrast to those grown at pH 8 (Treatment 3), even though the total amino acids were similar in these treatments. The cells exposed to 5% NaCl at 35 °C and pH 6 (Treatment 4) resulted in new different responses of the amino acids, with a decrease in the total average amino acid yield, produced by significant declines in aspartic acid, histidine, and lysine, while the levels of alanine and proline substantially increased relative to the control. In contrast, modifying the growth conditions to pH 8 at 35 °C and 5% NaCl (Treatment 5) resulted in no statistically significant alteration in the total average amino acid yield in comparison with the control but with significant alterations in the relative concentrations, with increases in glutamic acid, asparagine, and ornithine as well as decreases in histidine, proline, and methionine. It was thus established that the amino acid composition profiles of the cytoplasm differed significantly in response to growth under the various environmental conditions, with the main changes seen in aspartic acid, lysine, glutamic acid, histidine, and proline.

### Clustering Analysis

The datasets of all replicates collected from the control and treatment cells grown to stationary phases were further analyzed using heatmaps and hierarchical clustering to obtain an overview of the visual profiles of amino acid levels in all replicates (Figure 1). The analysis indicated consistent and reproducible patterns of amino acid levels for cells of both control and treatments. The heatmap cluster analysis clearly differentiates *S. aureus* amino acids based on treatment groups and demonstrates the reproducibility within treatment samples. The heatmap clustering was based on the Euclidean distance measure for similarity and the Ward clustering method. This statistical method clearly showed that the different conditions of growth resulted in various clusters on the basis of their amino acid compositions. The rows of the heatmap represent the amino acids, and the columns represent the samples whereas the color scale represents the relative fold change in each amino acid.

Supervised multivariate partial least squares discriminate analysis (PLS-DA) was performed to further characterize the differences in the amino acid compositions between the various treatment groups. The PLS-DA model was constructed after normalization was performed by the median, followed by log transformation and auto-scaling (mean-centered and divided by the standard deviation of each variable). The quality parameters of the PLS-DA model were as follows: R2 = 0.851 and Q2 = 0.695. Two-dimensional plots of the PLS-DA model established that the amino acid profiles of the treatment groups with 5% NaCl were most distinct from each other and those of the control samples, representing very different amino acid compositions (Figure 2a). However, Treatments 2 and 3, with no added salt, were positioned very close and adjacent, and Treatment 3 was indistinguishable from the reference control. Treatment 4, with pH 6 and 5% added NaCl, was separated far from all the other treatments, with substantial dispersal along the negative side of Component 1. Variable important projection (VIP) specified the contribution of the amino acids to the differentiation, and variables (amino acids) with values of >1 made a major contribution to discriminating among the different growth treatments in the PLS-DA model (Figure 2b). This analysis exhibited that the biological samples were closely grouped depending on the growth conditions, representing great reproducibility within samples defined by characteristic amino acid profile compositions based on the treatment.

To determine the amino acids that exclusively increased or decreased within each treatment van diagram analysis was applied. The analysis revealed that each treatment had at least one amino acid exclusively increased except for treatment 3 (Figure 3a), but only treatments 2 and 4 had unique decreased amino acids (Figure 3b). Treatment 4 had the highest number of altered amino acids, six and fourteen increased and decreased, respectively. Two of the altered amino acids in treatment 4 were exclusively increased including 4-hydroxyproline and glutamine whereas three amino acids were uniquely decreased including leucine, al-lo-Isoleucine and proline-hydroxyproline. Treatment 5 (pH 6 and 35 °C with 5% NaCl) which only differed from treatment 4 by altering pH 6 to 8, had no amino acids uniquely decreased, but two amino acids exclusively increased Including asparagine and ornithine. Proline-hydroxyproline was uniquely increased in treatment 1 and leucine was exclusively increased in treatment 2. The total number of non-overlapping circles reflects the total number of differentially regulated amino acids found only in that relevant experimental group; the overlapping parts of the circles indicate the differentially regulated amino acids common amongst the treatments.

## 4. Discussion

The outcomes of this investigation provided obvious evidence that *S. aureus* induced significant changes in its amino acid composition following growth to the stationary phase under altered conditions, including variations in temperature, osmotic pressure and pH that mimic a wound site. Amino acids were chosen, as they characterize important components of the cytoplasmic pool of metabolites due to their comprehensive roles in many biological processes, including protein synthesis, and specific amino acids are utilized for cell wall synthesis and nucleic acids, whereas others are readily used for energy substrates and as osmoprotectants.

The clustering heatmap and PLS-DA analyses of the dataset showed that slight adjustments in pH and temperature could induce differential alterations in the cytoplasmic amino acid compositions of *S. aureus* in response to the exposure to stress challenges (Figure 1 and Figure 2). It was thus suggested that *S. aureus* responded to fluctuations in various growth conditions by implementing the best metabolic response strategies for ideal survival in a specific dynamic setting. The particular amino acid responses to osmotic challenges were governed by other environmental conditions including temperature and pH levels. Under a similar range of pH and temperature, *S. aureus* had different metabolic responses to the hydrogen peroxide challenge [3]. These outcomes support the hypothesis that *S. aureus* promptly adapts to combinations of altered growth parameters by modifying its cytoplasmic amino acid composition to achieve optimal homeostasis related to the changes in the growth conditions, including those representative of the human wound site. Our previous work has shown that bacterial cells grown to the mid-exponential phase under altered environmental conditions resulted in different amino acid and protein compositions [21,23]. This was interpreted as a strategy used by bacterial cells to enhance adaptation processes and thus survivability. The detailed roles of the adjusted metabolic homeostasis in the current study are not clear at this stage, but comparative levels within the different patterns were steady between samples under a specific set of environmental conditions and were reproducible across repeated experiments. Cytoplasmic amino acid levels affected the physiological and virulence levels by stringently regulating the activity and responses of the nucleotide signaling systems, with subsequent pleiotropic influences, such as the increased expression of β-lactam resistance [27]. This particular adjustment in the amino acid levels may be driven by specific demands for stress-responsive protein synthesis, where modified groups of proteins would be vital under the different growth conditions. The profiles of amino acids also serve as snapshots that reflect the activity of the numerous metabolic pathways that are utilized in their biosynthesis.

The outcomes of the present investigation also suggested that the responses of the bacterial cells to changes in the environmental conditions could be determined by the combination of several parameters in the environmental conditions. There is an impressive capability in bacteria to sense and react to changes in the environmental parameters to regulate protein production and metabolic homeostasis [4,20,21]. The alterations in the cytoplasmic amino acid profiles under variations in the pH, osmotic pressure, and temperature could arguably correspond to different populations of bacteria under each treatment. The control and centroid experiments were repeated as two separate sets of three replicates across the experimental period, resulting in reproducible cytoplasmic amino acid profiles. Changes in any one or more of the environmental factors resulted in a distinct set of reproducible amino acid patterns, as previous studies demonstrated that the exposure of *S. aureus* to changes in the environment, measured via parameters, led to the formation of small colony variants (SCVs) on plate agars [28,29]. Another study has also reported that distinct phenotypes of *S. aureus* could be formed in response to a combination of environmental conditions similar to those used in the current study [30]. On this basis, we established the hypothesis that bacterial cells at the stationary phase are continually adjusting to stress-induced conditions by prompting mechanisms resulting in characteristic phenotypes that allow withstanding the modifications in the environmental conditions. These substantial modifications of the cytoplasmic amino acids at the stationary phase and under induced stress conditions were interpreted as involving an obligatory element of the cell acclimatization processes. Likewise, responses were observed in cytoplasmic and up-taken amino acids at the exponential phase following growth under various altered conditions [21,23]. This concept was also investigated in *Staphylococcus lugdunensis* following growth in liquid media using a similar experimental design, resulting in changes associated with fatty acids and cell size that were characteristic of the prevailing treatment conditions [31]. It was also found that *S. aureus* displayed a reduced cell size and increased resistance to acidic conditions in response to starvation [32,33].

## 5. Conclusions

The adaptation of *S. aureus* to various stress induced conditions is adequate and comprises numerous metabolite complexes, many of which show rapid changes in levels as the growth conditions are altered. This investigation concluded that the amino acid profiles of *S. aureus* were significantly altered in response to exposing the cells to combinations of potential stress conditions during growth, which might occur in the wound site. Moreover, certain changes in amino acid levels would offer an adaptation mechanism associated with the evolutionary survival of this bacterium. It was suggested that specific systematic strategies were instigated to enhance survivability in response to changes in temperature, osmolality, and pH to achieve optimal metabolic homeostasis via alterations to specific amino acids. The elucidation of the survival mechanisms of *S. aureus* under altered conditions representative of the human wound site is of great interest since it is a necessary measure for controlling bacterial survival and hence combating infections.

## Figures and Tables

**Figure 1 microorganisms-10-01503-f001:**
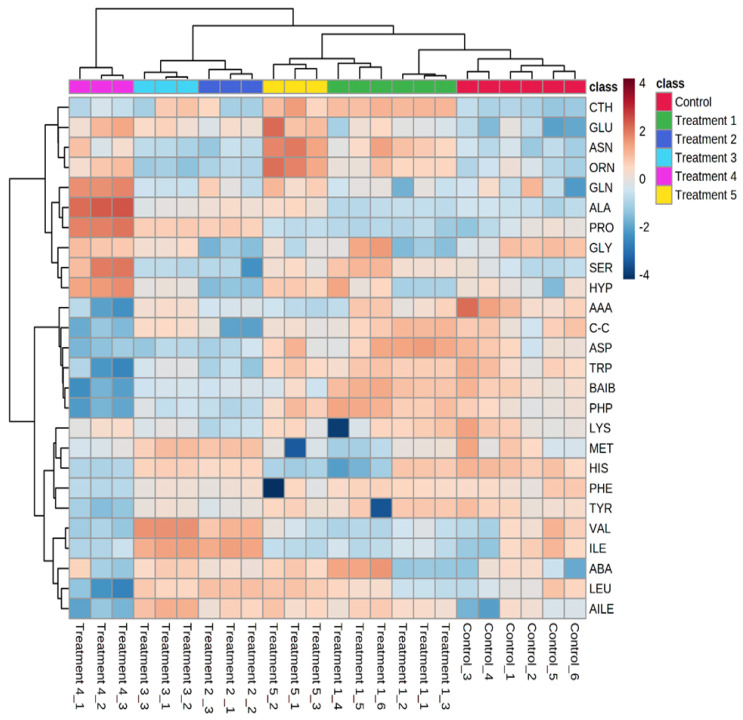
Heatmap exhibition of amino acid profiles from bacterial cells harvested at stationary phase from control and treatments samples (1–5). Each column denotes one biological sample; each row denotes one targeted amino acid. The colors represent relative amino acid abundance in each replicate.

**Figure 2 microorganisms-10-01503-f002:**
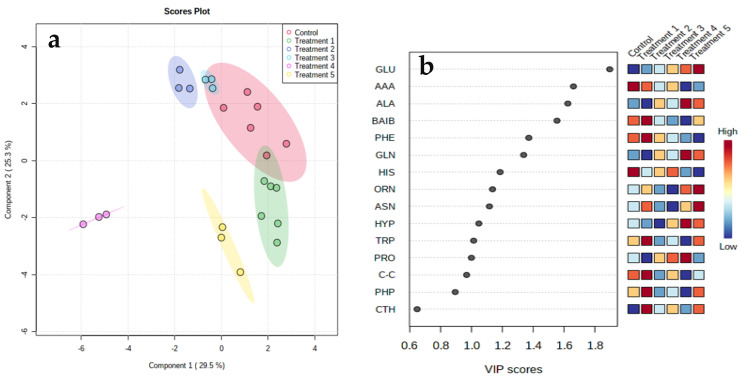
(**a**) Cross-validated PLS-DA score plot comparing the amino acid profiles of the clinical strain (red) and the ATCC29213 (green) strains, which shows the separation achieved during growth to the stationary phase. Colored dots represent individual replicates: 38.8% and 16.2% are the scores of Component 1 and Component 2, respectively, in the PLS-DA. (**b**) VIP plot exhibiting the 15 most important amino acids identified by PLS-DA for the cells grown to the stationary phase. Colored boxes on the right indicate the relative concentration of the corresponding amino acids in the samples. The VIP is the weighted sum of squares of the PLS-DA loading plot, taking the amount of the Y variable explained into account in each dimension.

**Figure 3 microorganisms-10-01503-f003:**
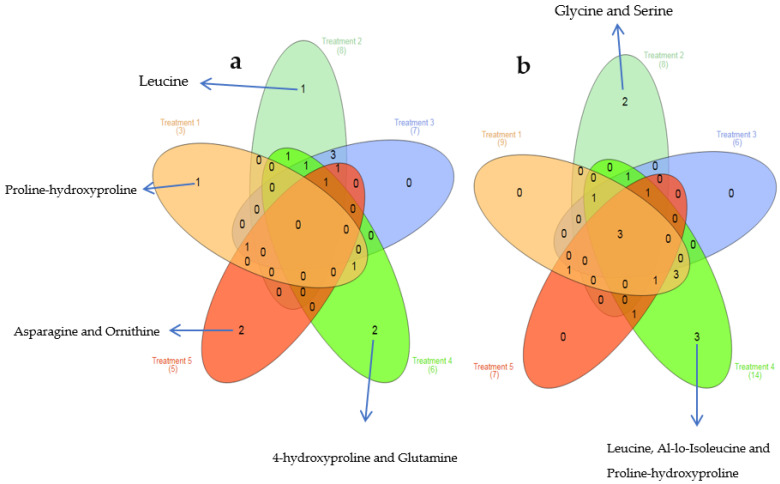
Venn diagram of the altered amino acids in all treatments. (**a**) Increased amino acids within all experimental groups. Unique increased amino acids were labelled next to each respective treatment (**b**) Indicates all amino acid that were decreased all treatments. Exclusive decreased amino acids were labelled beside respective treatment.

**Table 1 microorganisms-10-01503-t001:** The concentrations of *S. aureus* cytoplasmic amino acids following growth under differing experimental regimes. The concentration is expressed as nmol mg^−1^ dry cell weight (Mean + SD, *p* < 0.05).

Amino Acid Names	Amino Acids	Control (pH 7, 37 °C, 0% NaCl)	Treatment 1 (2.5% NaCl, pH 7, 37 °C)	Treatment 2 (pH 6, 35 °C)	Treatment 3 (pH 8, 35 °C)	Treatment 4 (pH6, 5% NaCl, 35 °C)	Treatment 5 (pH 8, 5% NaCl, 35 °C)
Alanine	ALA	3.38 ± 0.37	2.04 ± 0.15	7.96 ± 0.69 ^a^	5.45 ± 0.50	26.05 ± 4.35 ^a^	6.15 ± 0.40
Glycine	GLY	4.55 ±1.93	2.25 ± 1.64	1.44 ± 0.12 ^a^	3.74 ± 0.38	3.44 ± 0.96	2.23 ±0.71
α-Aminobutyric acid	ABA	0.20 ± 0.15	0.41 ± 0.42	0.37 ± 0.04	0.53 ± 0.08	0.14 ± 0.20	0.40 ± 0.04
Valine	VAL	6.46 ± 3.01	2.87 ± 0.28 ^a^	9.47 ± 0.46 ^a^	10.59 ± 0.50 ^a^	2.50 ± 0.54 ^a^	4.04 ± 0.44
ß-Aminoisobutyric acid	BAIB	1.62 ± 0.20	1.61 ± 0.25	0.84 ± 0.03 ^a^	0.72 ± 0.03 a	0.00 ± 0.00 ^a^	0.74 ± 0.19 ^a^
Leucine	LEU	2.23 ± 1.19	1.24 ± 0.43	3.77 ± 0.41 ^a^	2.84 ± 0.25	0.42 ± 0.23 ^a^	2.52 ± 0.36
Allo-Isoleucine	AILE	0.25 ± 0.18	0.32 ± 0.05	0.58 ± 0.05 ^a^	0.94 ± 0.11 ^a^	0.00 ± 0.00 ^a^	0.47 ± 0.11 ^a^
Isoleucine	ILE	1.76 ± 1.03	0.68 ± 0.11 ^a^	3.41 ± 0.24 ^a^	3.08 ± 0.20 ^a^	0.63 ± 0.15 ^a^	0.76 ± 0.12 ^a^
Serine	SER	1.23 ± 0.30	3.02 ± 1.32 ^a^	0.64 ± 0.46 ^a^	0.86 ± 0.06	9.32 ± 3.28 ^a^	2.12 ± 0.20
Proline	PRO	10.83 ± 6.77	3.08 ± 0.55 ^a^	32.01 ± 2.99 ^a^	30.14 ± 2.30 ^a^	79.79 ± 17.70 ^a^	4.98 ± 1.01 ^a^
Asparagine	ASN	4.84 ± 0.97	4.59 ± 0.42	5.40 ± 0.13	4.87 ± 0.21	4.53 ± 1.64	7.72 ± 0.90 ^a^
Aspartic acid	ASP	180.4 ± 32.5	156.63 ± 29.33	124.78 ± 10.18 ^a^	102.24 ± 9.08 ^a^	55.51 ± 11.06 ^a^	158.01 ± 16.14
Methionine	MET	6.14 ± 1.83	2.31 ± 0.84 ^a^	10.36 ± 0.86 ^a^	9.38 ± 1.08 ^a^	2.63 ± 0.55 ^a^	2.43 ± 1.74 ^a^
4-Hydroxyproline	HYP	0.47 ± 0.28	0.62 ± 0.98	0.00 ± 0.00	0.49 ± 0.04	3.12 ± 0.72 ^a^	1.07 ± 0.05
Glutamic acid	GLU	51.94 ±13.87	55.82 ± 9.98	110.07 ± 2.72 ^a^	116.94 ± 4.25 ^a^	95.62 ± 16.66 ^a^	152.76 ± 40.73 ^a^
Phenylalanine	PHE	3.26 ± 1.20	2.10 ± 0.14	3.01 ± 0.15	2.66 ± 0.17	0.99 ± 0.26 ^a^	1.51 ± 1.07 ^a^
α-Aminoadipic acid	AAA	3.31 ± 1.73	1.09 ± 0.43 ^a^	0.93 ± 0.05 ^a^	1.57 ± 0.08 ^a^	0.15 ± 0.21 ^a^	0.46 ± 0.07 ^a^
Glutamine	GLN	2.11 ± 2.73	0.67 ± 0.35	2.72 ± 2.30	3.28 ± 3.62	13.25 ± 3.39 ^a^	3.96 ± 1.55
Ornithine	ORN	5.53 ± 1.14	5.73 ± 0.58	5.47± 0.39	4.12 ± 0.13	6.58 ± 1.06	13.21 ± 0.91 ^a^
Lysine	LYS	67.22 ± 13.05	36.24 ± 17.54 ^a^	34.22 ± 1.69 ^a^	43.96 ± 1.62 ^a^	36.67 ± 5.32 ^a^	47.38 ± 1.87 ^a^
Histidine	HIS	37.54 ± 10.13	10.82 ± 10.34 ^a^	23.19 ± 0.93 ^a^	26.18 ± 1.75 ^a^	1.48 ± 0.40 ^a^	1.53 ± 0.39 ^a^
Tyrosine	TYR	5.47 ± 0.71	3.55 ± 1.58 ^a^	4.00 ± 0.13	4.28 ± 0.25	0.93 ± 0.36 ^a^	4.35 ± 0.13
Proline-hydroxyproline (dipeptide)	PHP	1.85 ± 0.40	3.14 ± 1.12 ^a^	0.66 ± 0.15	0.89 ± 0.25	0.00 ± 0.00 a	2.59 ± 0.77
Tryptophan	TRP	0.99 ± 0.30	0.65 ± 0.07 ^a^	0.26 ± 0.07 ^a^	0.45 ± 0.05 ^a^	0.07 ± 0.10 ^a^	0.77 ± 0.09
Cystathionine	CTH	0.00 ± 0.00	0.21 ± 0.01 ^a^	0.08 ±0.11	0.17 ± 0.12 ^a^	0.00 ± 0.00	0.25 ± 0.07 ^a^
Cystine	C-C	0.42 ± 0.20	0.34 ± 0.10	0.10 ± 0.15 ^a^	0.37 ± 0.01	0.00 ± 0.00 ^a^	0.21 ± 0.01
Total Amino acids	Total AA	407.01 ± 47.9	301.67 ± 79.03 ^a^	385.64 ± 25.48	380.36 ± 27.09	343.81 ± 60.20	422.63 ± 28.26

^a^ significantly altered amino acids (*p* < 0.05).

## Data Availability

Not applicable.

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
