# Peer review of "Changes in Amino Acid Metabolism of Staphylococcus aureus following Growth to the Stationary Phase under Adjusted Growth Conditions"

_microorganisms, 2022, doi:10.3390/microorganisms10081503_

Round 1
Reviewer 1 Report
The authors present a set of metabolic changes that occur in different in-vitro conditions representative of wound sites that could serve as an infection point for S.aureus and potentially be used for S.aureus survival.
I find the information presented is rather cumbersome to follow and adds minimally to the existing knowledge on wound site infections of S.aureus and the linked metabolism. This area has significantly benefited from the work done by the group of authors, I however find this paper to be lacking.
In terms of control experiments:
I would like to see amino acid profiling in pH 6, 7, and 8 to be done at all the same temperature either 37C or 35C.
Similarly no mention of the impact of osmolarity change is represented in the treatment conditions 1 4 and 5.
Whereby, does the addition of a different salt lead to similar impact on amino acid metabolism. Is the change in amino acids content proportional to the increasing osmolarity of the growth media?
I recommend the authors to add a pH 6 at 37 and pH 8 at 37C condition and then discuss the impact of pH and temperature separately. Changing two conditions seems like an odd choice when trying to elucidate the minimally changing metabolic response.
I understand the need to be transparent with all amino acid changes, however, the Table 1 is cumbersome, I recommend the authors to add only significantly changing amino acids. The other information can be either listed as NS or in a supplementary table.
I am also unsure of what the units for concentration are in this table? Are the normalized to total dry cell weight/ OD on harvest? I am considering that even at stationary phase there will be differences in the total cell volume.
Similarly, I would like the authors to comment on if the amino acid profile in the control condition are proportionally similar to all the other times they have measured the amino acid profile.
Either the amino acid abbreviations listed or are they accepted abbreviations for the journal?
In Figure 1 the heat map needs to discuss why there is a variable number of replicates for Control and Treatment 1 vs the others?
If so, did the authors use the appropriate tailed t-test for the statistical testing?
Also, was any multiple hypothesis correction( bonferroni/ Benjamini Hochberg) performed for all the p<0.05 listed in Table 1?
Since the PLS-DA model represents multiple groups I would recommend the authors perform something called a Mahalanobis distance dendrogram to show if the clustering is significant. the relative distances are a poor assessment of variation between groups in such multiple group PLS-DA modeling.
I also recommend the authors perform a CV-Anova cross validation for the PLS-DA even if they are using only Vip features out of the model.
For Figure 3 I think the authors could represent the information in a more meaningful way by adding the metabolites each of the Venn diagram nodes represent, that will assist the readers in following along.
Lastly, since the authors have performed similar analyses a few different ways and reported the information linked to those metabolic changes, adding a figure that summarized how key metabolites are altered in high salt, high pH and low pH conditions on a pathway map representing s.aureus metabolism will further enhance the manuscript and provide the audience with a summary of what these metabolic profiles look like and how / which set of enzyme could be used to regulate the growth and infection of S.aureus in such wound healing circumstances.
Without this, the paper seems redundant.
Author Response
Dear editor of microorganisms Journal,
We, the authors of microorganisms-1802697, appreciate reviewer 1 comments and suggestions which helped us to revise the manuscript. Most comments seemed to be fair and reasonable, so we paid heed to their advice and suggestions, and the manuscript has been accordingly revised. We believe that the contents and the clarity of our paper are much improved in the revised version. Below are point-by-point responses for the reviewer comments. Finally, a clean revised manuscript and revised manuscript with “Track Changes” have also been uploaded alongside this document.
Thank you again for consideration of our revised manuscript.
Reviewer 1
Comments and Suggestions for Authors
The authors present a set of metabolic changes that occur in different in-vitro conditions representative of wound sites that could serve as an infection point for S.aureus and potentially be used for S.aureus survival.
I find the information presented is rather cumbersome to follow and adds minimally to the existing knowledge on wound site infections of S.aureus and the linked metabolism. This area has significantly benefited from the work done by the group of authors, I however find this paper to be lacking.
We acknowledge that the quantity of information and results presented is large. However, the evolving capacity to measure changes in metabolism and homeostasis via metabolic profiling enables a new insight to subtle alterations in cellular homeostasis. The advanced statistical approaches enable exciting new ways to conceive and visualize interactions and changes.
In terms of control experiments:
- I would like to see amino acid profiling in pH 6, 7, and 8 to be done at all the same temperature either 37C or 35C.
- The study was designed to mimic potential conditions present on skin or within a wound site. Thus this study did not design to examine the pH response in aureus, but to assess the alterations in cytoplasmic amino acids under a various of environmental conditions to determine whether the cellular responses of exposure to an environmental parameter would be differed under a different set of environmental factors such as pH, temp, and osmotic pressure. Thus pH7 at 37 °C and no salt added was considered as the optimal conditions and used as reference control to compare with other experiments. Due to time constrain and logistic of the experiments, we were be able to investigate the following conditions pH6-7 , temperature 35-37°C and NaCl 0-5% added. However, in the future will consider the full factorial design to encompass the conditions suggested, but this would make consideration of the results even more “cumbersome”.
Similarly no mention of the impact of osmolarity change is represented in the treatment conditions 1 4 and 5.
- The results of treatment 1 are mentioned at lines 147 – 155
“Several specific changes in amino acid levels associated with different treatments were apparent, as shown in Table 1. The total average amino acid yield for the control samples was 407.01 ± 47.9 nmol mg-1 (mean ± SD), but it was significantly reduced in the bacterial cells exposed to 2.5 NaCl at pH 7 and 37 °C (Treatment 1). This reduction was mainly accomplished by 30–50% declines in histidine and lysine. Other treatments showed no significant changes in their total amino acids relative to the control samples, but the relative abundances of individual amino acids were significantly different in the treatments in comparison with the control, as noted in Table 1.”
- The results of treatments 4 and 5 were mentioned together with treatments 2 and 3 in the result section lines 157-169
“The growth of S. aureus at a lower temperature of 35 °C with no addition of sodium chloride at pH 6 (Treatment 2) resulted in extraordinarily different responses within individual amino acids in contrast to those grown at pH 8 (Treatment 3), even though the total amino acids were similar in these treatments. The cells exposed to 5% NaCl at 35 °C and pH 6 (Treatment 4) resulted in new different responses of the amino acids, with a decrease in the total average amino acid yield, produced by significant declines in aspartic acid, histidine, and lysine, while the levels of alanine and proline substantially increased relative to the control. In contrast, modifying the growth conditions to pH 8 at 35 °C and 5% NaCl (Treatment 5) resulted in no statistically significant alteration in the total average amino acid yield in comparison with the control but with significant alterations in the relative concentrations, with increases in glutamic acid, asparagine, and ornithine as well as decreases in histidine, proline, and methionine. It was thus established that the amino acid composition profiles of the cytoplasm differed significantly in response to growth under the various environmental conditions, with the main changes seen in aspartic acid, lysine, glutamic acid, histidine, and proline.”
Whereby, does the addition of a different salt lead to similar impact on amino acid metabolism. Is the change in amino acids content proportional to the increasing osmolarity of the growth media?
We did not set out to investigate the osmoprotectant response in S aureus. We sought to assess whether different environmental sets of parameters led to differential amino acid homeostasis in the cytoplasm. This is clearly outlined in lines 71 – 79. These results are extremely interesting in the broader context that S aureus achieves a differential homeostasis for each set of environmental conditions tested. As explained above, phenomena observed at pH 7 do not necessarily reflect what will happen under different environmental conditions with different pH. This study has shown new evidence that each set of environmental conditions elicited substantial and unique alterations in cytoplasmic amino acids. This study significantly supported the hypothesis that the bacterium was continually responding to the dynamic environment by modifying amino acid levels and thus optimizing metabolic homeostasis.
I recommend the authors to add a pH 6 at 37 and pH 8 at 37C condition and then discuss the impact of pH and temperature separately. Changing two conditions seems like an odd choice when trying to elucidate the minimally changing metabolic response.
Wound site is an avenue of complex conditions, so to mimic these conditions we had to design experiments with an array of environmental conditions. We were not setting out to investigate the impacts of changing one parameter such as pH alone whilst holding everything else constant, because this is not how it happens in the wound site. One out come from this study is that a different salt level leads to a different response to pH. Or different temperatures alter the way in which cells respond to salt or pH. The general outcome is that cells adopt the best metabolic homeostasis that will allow survival in the present conditions.
I understand the need to be transparent with all amino acid changes, however, the Table 1 is cumbersome, I recommend the authors to add only significantly changing amino acids. The other information can be either listed as NS or in a supplementary table.
We have certainly adopted this approach for presentation of our proteomics papers. However, working in metabolomics, it is important to provide a sense of accuracy for the measures and the standard error associated with estimates of the mean values. For people working with this investigation as a future reference, these values will be immensely helpful to understand the magnitudes of the changes in amin acid homeostasis for each of the cytoplasmic amino acids. It will also allow future workers to investigate similar design matrices and study effluxes of amino acids from the cells under certain conditions. Future work will also look at the changes that occur during growth of the cells in culture as the environmental conditions change with nutrient losses and buildup of toxic wastes. It will be useful for researchers to be able to compare the changes.
- Full name of each amino acid was added to the Table 1 as suggested by reviewer 1
I am also unsure of what the units for concentration are in this table? Are the normalized to total dry cell weight/ OD on harvest? I am considering that even at stationary phase there will be differences in the total cell volume.
Amino acid concentrations were calculated as nmol mg−1 dry cell weight (Mean + SD, P < 0.05).
Similarly, I would like the authors to comment on if the amino acid profile in the control condition are proportionally similar to all the other times they have measured the amino acid profile.
The control and centroid samples were assessed on two separate occasions throughout the experimental period with little variance as shown by the tight clustering of the combined sample sets in the PLS-DA and heatmap figures.
Either the amino acid abbreviations listed or are they accepted abbreviations for the journal?
The full name of each amino acid was added in the table 1 as suggested by the reviewer 1. Many thanks.
In Figure 1 the heat map needs to discuss why there is a variable number of replicates for Control and Treatment 1 vs the others?
The experiments of Control and treatment 1 were done two separate times to control any temporal variations. See lines 102 – 104.
If so, did the authors use the appropriate tailed t-test for the statistical testing?
We used Post-hoc Duncan’s test analysis to determine the amino acids that were significantly altered in response to various environmental conditions
Also, was any multiple hypothesis correction( bonferroni/ Benjamini Hochberg) performed for all the p<0.05 listed in Table 1?
The Duncan’ test was performed to cater for this effect.
Since the PLS-DA model represents multiple groups I would recommend the authors perform something called a Mahalanobis distance dendrogram to show if the clustering is significant. the relative distances are a poor assessment of variation between groups in such multiple group PLS-DA modeling.
We used three different analyses including PLS-DA, Heatmap and Euclidean distance dendrogram to asses our data and all these analyses showed very similar outcomes indicating that all replicates of a treatment were very well clustered, yet separated in a different degree from control and other treatments. Both PLS-DA and Heatmap were present in the manuscript and Euclidean distance dendrogram is shown below.
I also recommend the authors perform a CV-Anova cross validation for the PLS-DA even if they are using only Vip features out of the model.
The validation of PLS-DA was checked through cross validation. The analysis produced 8 components and three of which were significant. Cross-validation including R2 and Q2 values were used to evaluate the goodness of fit and predictive ability of the PLS-DA model. The quality parameters of the PLS-DA model were as follows: R2 = 0.851 and Q2 = 0.695. Please see line 203-204.
“Q2 is an estimate of the predictive ability of the model, and is calculated via cross-validation (CV). In each CV, the predicted data are compared with the original data, and the sum of squared errors is calculated. The prediction error is then summed over all samples (Predicted Residual Sum of Squares or PRESS). For convenience, the PRESS is divided by the initial sum of squares and subtracted from 1 to resemble the scale of the R2. Good predictions will have low PRESS or high Q2. It is possible to have negative Q2, which means that your model is not at all predictive or is overfitted. For more details, refer to an excellent paper by (SzymaÅ„ska, et al)”.
For Figure 3 I think the authors could represent the information in a more meaningful way by adding the metabolites each of the Venn diagram nodes represent, that will assist the readers in following along.
Would like to thank the reviewer for this suggestion, the amino acids were added to the Venn diagram
Lastly, since the authors have performed similar analyses a few different ways and reported the information linked to those metabolic changes, adding a figure that summarized how key metabolites are altered in high salt, high pH and low pH conditions on a pathway map representing s.aureus metabolism will further enhance the manuscript and provide the audience with a summary of what these metabolic profiles look like and how / which set of enzyme could be used to regulate the growth and infection of S.aureus in such wound healing circumstances.
Without this, the paper seems redundant.
The focus is not about identifying which metabolites are linked to alterations in high salt, high pH and low pH or changes in temperature. The key feature is that there are obvious interactions between the environmental parameters which alter how the cell responds. It is thus irrelevant to try to align metabolites to pH changes or temperature changes, because these responses will vary depending on osmolarity. The hypothesis and aim from 71 – 79 clearly address this feature. The outcome of the investigation is that each set of environmental parameters will dictate a specific cytoplasmic homeostasis. This specific homeostasis is reproducible when grown on identical conditions. This is a major leap in understanding a new concept.
“It was hypothesized that S. aureus induces significant changes in its amino acid com-positions to obtain optimal metabolic homeostasis and thus enhance survivability during exposure to stressful conditions present in the stationary phase and during exposure to combinations of environmental parameters representative of human wound site, including variations in temperature, pH and osmotic pressure. This study aimed to explore the variations in amino acid compositions following the growth of bacterial cells to stationary phase under variations in temperature, pH and sodium chloride (NaCl) representative of a wound site.”

Reviewer 2 Report
The manuscript of Alreshidi et al. analyzes differences in amino acid concentrations in S. aureus exposed to different conc. of NaCl, pH in temperatures. The authors claim that these are conditions that S. aureus encounter in the human wound site. I think that this physiological response to changes in the environmental conditions is interesting to study; however, I see some significant problems with this study:
- The stressor in the environment may resemble the conditions, but putting this in the title is over-estimation
- This is a response of a single strain. In serious conclusion, more strains have to be studied.
- How was the bacterial growth measured, and how was the stationary phase determined?
- The S. aureus must be written in italic throughout the entire text.
- In the discussion, changes in the amino acid concentrations have to be more precisely correlated to specific metabolic pathways.
Author Response
Dear editor of microorganisms Journal,
We, the authors of microorganisms-1802697, appreciate the reviewer 2 comments and suggestions which helped us to revise the manuscript. Most comments seemed to be fair and reasonable, so we paid heed to their advice and suggestions, and the manuscript has been accordingly revised. We believe that the contents and the clarity of our paper are much improved in the revised version. Below are point-by-point responses for the reviewer comments. Finally, a clean revised manuscript and revised manuscript with “Track Changes” have also been uploaded alongside this document.
Thank you again for consideration of our revised manuscript.
Reviewer 2
Comments and Suggestions for Authors
The manuscript of Alreshidi et al. analyzes differences in amino acid concentrations in S. aureus exposed to different conc. of NaCl, pH in temperatures. The authors claim that these are conditions that S. aureus encounter in the human wound site. I think that this physiological response to changes in the environmental conditions is interesting to study; however, I see some significant problems with this study:
- The stressor in the environment may resemble the conditions, but putting this in the title is over-estimation
The title has been modified based on the reviewer’ suggestion
- This is a response of a single strain. In serious conclusion, more strains have to be studied.
Fair comment, but you have to start somewhere! Future work can focus on exploring these cellular responses in different strains and species.
- How was the bacterial growth measured, and how was the stationary phase determined?
The text has been modified to indicate how the bacterial growth measured and how was the stationary phase determined.
The experimental cultures were prepared by transferring five ml of overnight culture to 95 ml of TSB culture media (OD 600 =0.1 ≈ 108 CFU/ml) and grown to stationary phase with designated growth conditions prior the analysis of amino acid compositions. Growth was monitored by aseptically checking the absorbance at 600nm to estimate cell numbers, so that the cultures could be harvested at stationary phase. Cultures were harvested by centrifugation at 6000 xg for 20 min followed by washing three times with phosphate buffer saline (PBS).The washed cells were directly quenched using liquid nitrogen and placed in freezer-dryer machine and following extraction for metabolic profiling analysis. The experiments of control and treatment 1) pH7 at 37°C with NaCl added (2.5%) were repeated two separate times to optimize statistical assessment and ensure the reproducibility of the experiments.
- The S. aureus must be written in italic throughout the entire text.
- aureus is written in italic throughout the entire text.
- In the discussion, changes in the amino acid concentrations have to be more precisely correlated to specific metabolic pathways.
The focus for this study was not to try to identify the specific metabolic pathways involved in response to alterations in high salt, high pH and low pH or changes in temperature. This would make it a huge voluminous work and this investigation represents the first step in this process.
The key feature is that there are obvious interactions between the environmental parameters which alter how the cell responds. It is thus irrelevant to try to align metabolites to pH changes or temperature changes, because these responses will vary depending on osmolarity. The hypothesis and aim from 71 – 79 clearly address this feature. The outcome of the investigation is that each set of environmental parameters will dictate a specific cytoplasmic homeostasis. This specific homeostasis is reproducible when grown on identical conditions. This is a major leap in understanding and a new concept.

Round 2
Reviewer 1 Report
The authors made some of the changes I requested, and have provided an explanation for others.
Author Response
Dear Prof. Alina Lavinia Avram, editor, microorganisms
Subject: Responses to editors’ comments. Manuscript ID: microorganisms-1802697, entitled “Changes in Amino acid metabolism of Staphylococcus aureus following growth to the stationary phase under adjusted growth conditions””
We would like to thank you for your time for providing the comments and suggestions to make this manuscript more scientifically sound. The comments and suggestions have been sincerely looked, considered and manuscript is revised accordingly. The responses are given in a point-by-point manner. A clean revised manuscript and revised manuscript with “Track Changes” have also been uploaded alongside this document.
Thank you again for consideration of our revised manuscript.
I would like to thank reviewers for the careful evaluation. After reading the reviewer comments, I also agree with them and kindly ask the authors should improve some points:
- The authors should clarify the novelty of the current study with your previous publication.
M Alreshidi M, Dunstan RH, M Macdonald M, K Singh V, K Roberts T. Analysis of Cytoplasmic and Secreted Proteins of Staphylococcus aureus Revealed Adaptive Metabolic Homeostasis in Response to Changes in the Environmental Conditions Representative of the Human Wound Site. Microorganisms. 2020 Jul 20;8(7):1082. doi: 10.3390/microorganisms8071082. PMID: 32698515; PMCID: PMC7409162.
The present investigation is totally different from the previous published paper mentioned above. The current investigation looked at the changes in the cytoplasmic amino acid metabolites of the cells grown to the stationary phase under various growth conditions similar to those present in a wound site. It thus represents the changes in amino acid homeostasis in the cells grown under different growth conditions. In contrast, the published study investigated the alterations in cytoplasmic and secreted in proteins of the cells grown to mid-exponential phase under similar conditions used in this study. This earlier study was a proteomic investigation and considered both cytoplasmic proteins and those excreted into the external medium.
- Since the study was designed to mimic potential conditions present on skin or within a wound site, it would be better to have an illustrated figure presenting the location of skin and wound sites which present for 5 treatment conditions in the study. It would enhance the clarity for the aim of the study, otherwise the readers will be confused.
The wound site is a complex structure with varying conditions depending on the location within the wound site. In addition, the environmental conditions vary depending on the age of the wound, the state of repair, the extent of damage, the inflammation response and the presence or absence of infectious bacteria. This is described in lines 59 – 65.
The range of conditions selected for the study were taken from the literature reports of the ranges measured in terms of pH, temperature and osmolarity taken at various times post-wounding.
As such, the 5 treatment conditions represent combinations of selected conditions within the reported ranges of the three major variable parameters. It is thus likely that that these conditions could arise within the wound site at a given time and it is also possible that multiple conditions may be present in micro-environments within the wound site depending on location.
It would be extremely difficult to develop a diagram that would convey this in a simplistic manner.
- In line 229-239: The result from figure 3 should be clarified in more detail with the mention of the name of amino acids which are increased and decreased.
The manuscript is now modified based on the above suggestion
- In the corrected Figure 3, the amino acids were added however the quality is bad, the amino acids were inserted on the number, letters were small to follow, lacking amino acids in Fig 3b- treatment 2. The figure legend should explain in more detail that the readers can follow the figure.
- Really appreciated it this suggestion. We removed the amino acids from the diagram to avoid interference with numbers. We stated these amino acids in the text as suggested in the comment above and next to each respective treatment.
- - Table 1 is oversize - some letters on the right side are missing.
The table is adjusted to fit in the page

Reviewer 2 Report
The manuscript has been corrected accordingly.
Author Response

(The authors gave the same response as above.)
